# Multi-Drug Resistant *Staphylococcus aureus* Carriage in Abattoir Workers in Busia, Kenya

**DOI:** 10.3390/antibiotics11121726

**Published:** 2022-12-01

**Authors:** Benear Apollo Obanda, Cheryl L. Gibbons, Eric M. Fèvre, Lilly Bebora, George Gitao, William Ogara, Shu-Hua Wang, Wondwossen Gebreyes, Ronald Ngetich, Beth Blane, Francesc Coll, Ewan M. Harrison, Samuel Kariuki, Sharon J. Peacock, Elizabeth A. J. Cook

**Affiliations:** 1Department of Veterinary Pathology, Microbiology and Parasitology, University of Nairobi, Nairobi P.O. Box 29053-00625, Kenya; 2Global One Health Initiative, The Ohio State University, Columbus, OH 43210, USA; 3Centre for Microbiology Research Nairobi, Kenya Medical Research Institute, Nairobi P.O. Box 54840-00200, Kenya; 4Public Health Scotland, Glasgow G2 6QE, UK; 5Institute of Infection, Veterinary & Ecological Sciences, Leahurst Campus, University of Liverpool, Chester High Road, Neston CH64 7TE, UK; 6International Livestock Research Institute, Nairobi P.O. Box 30709-00100, Kenya; 7Department of Public Health Pharmacology and Toxicology, University of Nairobi, Nairobi P.O. Box 29053-00625, Kenya; 8Division of Infectious Disease, Department of Internal Medicine, The Ohio State University, Columbus, OH 43210, USA; 9Department of Veterinary Preventive Medicine, College of Veterinary Medicine, The Ohio State University, Columbus, OH 43210, USA; 10Department of Medicine, University of Cambridge, Cambridge CB2 0QQ, UK; 11London School of Hygiene and Tropical Medicine, London WC1E 7HT, UK; 12Wellcome Sanger Institute, Hinxton CB10 1SA, UK; 13Department of Public Health and Primary Care, University of Cambridge, Cambridge CB1 8RN, UK

**Keywords:** *S. aureus*, MSSA, MRSA, abattoir, slaughterhouse, Kenya, HIV, AMR, antimicrobial resistance

## Abstract

Abattoir workers have been identified as high-risk for livestock-associated *Staphylococcus aureus* carriage. This study investigated *S. aureus* carriage in abattoir workers in Western Kenya. Nasal swabs were collected once from participants between February-November 2012. *S. aureus* was isolated using bacterial culture and antibiotic susceptibility testing performed using the VITEK 2 instrument and disc diffusion methods. Isolates underwent whole genome sequencing and Multi Locus Sequence Types were derived from these data. *S. aureus* (*n* = 126) was isolated from 118/737 (16.0%) participants. Carriage was higher in HIV-positive (24/89, 27.0%) than HIV–negative participants (94/648, 14.5%; *p* = 0.003). There were 23 sequence types (STs) identified, and half of the isolates were ST152 (34.1%) or ST8 (15.1%). Many isolates carried the Panton-Valentine leucocidin toxin gene (42.9%). Only three isolates were methicillin resistant *S. aureus* (MRSA) (3/126, 2.4%) and the prevalence of MRSA carriage was 0.4% (3/737). All MRSA were ST88. Isolates from HIV-positive participants (37.0%) were more frequently resistant to sulfamethoxazole/trimethoprim compared to isolates from HIV-negative participants (6.1%; *p* < 0.001). Similarly, trimethoprim resistance genes were more frequently detected in isolates from HIV-positive (81.5%) compared to HIV-negative participants (60.6%; *p* = 0.044). *S. aureus* in abattoir workers were representative of major sequence types in Africa, with a high proportion being toxigenic isolates. HIV-positive individuals were more frequently colonized by antimicrobial resistant *S. aureus* which may be explained by prophylactic antimicrobial use.

## 1. Introduction

*Staphylococcus aureus* is a common commensal of the skin, thought to persistently colonize approximately one third of the human population [1]. Nasal carriage of *S. aureus* is a recognized risk factor for skin and soft tissue infections (SSTI) in the clinical setting [2,3]. *S. aureus* may also cause food poisoning and more serious conditions such as pneumonia, endocarditis, osteomyelitis, sepsis, and toxic shock syndrome [4]. HIV-positive individuals are more likely to be colonized by *S. aureus*, which accounts for significant morbidity in this group compared to the general population [5].

The sub-Saharan region is recognized as the world’s epicenter of the HIV/AIDS epidemic. The prevalence of HIV in Kenya is 5% [6]. The Kenyan Ministry of Health recommends the use of sulfamethoxazole/trimethoprim for the management of opportunistic infections in all HIV positive patients regardless of immunological status [7]. The prophylactic use of sulfamethoxazole/trimethoprim in HIV-positive individuals results in significant protection from a range of pathogens including *Toxoplasma gondii*, *Salmonella* sp., *Haemophilus* sp., *Staphylococcus* sp., and *Pneumocystis jiroveci* [8]. However, prophylactic use of sulfamethoxazole/trimethoprim in HIV-positive individuals in the sub-Saharan region has led to the emergence of antibiotic resistant and multidrug resistant *S. aureus* strains [9,10].

Antimicrobial resistance (AMR) poses a threat to life since infections caused by multi-drug resistant organisms have fewer treatment options available [11]. This is particularly important for the HIV-positive population. Injudicious use of antibiotics in human medicine, coupled with extensive antibiotic use in livestock production for both therapeutic and non-therapeutic reasons has led to the development of antibiotic resistant bacteria in people and animals [12,13]. Multi-drug resistant methicillin sensitive *S. aureus* (MSSA) and methicillin resistant *S. aureus* (MRSA) have been detected in animals and meat products [14]. There has also been documented transmission to livestock keepers and abattoir workers [15,16]. This implies that, abattoir workers’ nares can be colonized by *S. aureus* from contaminated meat, transforming them into carriers or reservoirs of *S. aureus.* Carriers can transmit the bacteria from their noses to other body parts, to the general population, or contaminate foods and food surfaces during handling [17]. These transmission routes have been reported in several European countries [18,19].

Multi-locus sequence typing is conducted on *S. aureus* to understand the molecular epidemiology of the isolates including the evolution, source attribution and transmission through the sequencing of seven housekeeping genes [20]. Clonal complexes (CC) describe *S. aureus* lineages by grouping sequence types (STs) where at least 5/7 alleles are identical between STs in the group [21]. The dominant STs vary between countries and within countries depending on the source. The dominant MSSA STs in Africa are ST5, ST8, ST15, ST30, ST121, ST152 and the dominant MRSA STs are ST5, ST8, ST80, ST88, ST239/ST241 [22]. There is limited information of the predominant sequence types in Kenya but there are increased reports of ST5, ST8, ST22, with the predominant MRSA ST being ST239/241 [23,24,25,26,27,28].

Isolates of *S. aureus* may carry genes for virulence factors such as Panton-Valentine leukocidin (PVL), and the toxic shock syndrome toxin (TSST-1) [29,30,31]. PVL is a virulence factor which is associated with SSTI and has a debatable role in causing necrotizing pneumonia [32,33,34,35] whereas TSST-1 results in toxic shock syndrome leading to lethal hypotension [29]. In sub-Saharan Africa *S. aureus* isolates more frequently carry the *pvl* gene with the median prevalence of *pvl*-positive MRSA being 33% (range from 0 to 77%; *n* = 15), compared to Europe where less than 5% of *S. aureus* isolates carry the *pvl* gene [23,36,37]. There is very little information about the prevalence of *tsst-1* gene carriage in Africa with one study in Nigeria reporting carriage in human isolates to be 16% [38].

Here, we report on a study of MSSA/MRSA nasal carriage of abattoir workers in rural abattoirs in Busia County, western Kenya. This study aimed to establish the prevalence of MSSA and MRSA colonization and describe genetic characteristics of isolates obtained from abattoir workers in order to understand the epidemiology of *S. aureus* in this population of workers exposed to livestock. This was done by investigating the prevalence of *pvl* and *tsst-1* genes, antimicrobial susceptibility and diversity of STs. Due to the high proportion of HIV-positive individuals in this population we also endeavored to use this dataset to understand the effect of prophylactic use of sulfamethoxazole/trimethoprim in HIV-positive individuals on the emergence of MSSA/MRSA antibiotic resistance and associated virulence by comparing HIV-positive and HIV-negative workers. Genotypic characterization of virulent strains and their antibiotic resistance profiles will contribute to understanding the potential sources and transmission routes of *S. aureus* in this setting. This information will be valuable to Kenya’s National Policy for the prevention and containment of AMR [39].

## 2. Results

### 2.1. Description of Study Population

A total of 737/738 abattoir workers, recruited between February and November 2012, consented to a blood sample and a single nasal swab which was cultured for the presence of *S. aureus*. One participant declined to provide a blood sample and was excluded.

The majority of participants were men 711/737 (96.5%) and the mean age was 39 (range 18–82 years). The number of participants who tested positive for HIV was 89/737 (12.1%, 95%CI 9.9–14.7%). Additionally, 127/737 (17.2%) of participants had taken antibiotics in the previous month.

### 2.2. Prevalence of MRSA and MSSA among HIV-Positive and HIV-Negative Participants

*S. aureus* was isolated from 118/737 (16.0%; 95%CI 13.6–18.8%) participants. From 118 positive samples, 126 isolates were cultured in total since four participants had two separate strains and a further two participants had three strains identified from the same sample. Three isolates were MRSA and 123 isolates were MSSA, giving a prevalence of MRSA carriage of 0.4% (95%CI 0.1–1.2%), and MSSA carriage of 15.6 % (95%CI 13.2–18.5%), respectively (Table 1). There were no known relationships between the three MRSA carriers, and individuals worked at different abattoirs.

Of the HIV-positive workers, 24/89 were positive for *S. aureus* (27.0% (95%CI 19.1–36.7%). In contrast, 94/648 HIV-negative workers were positive for *S. aureus* (14.5%, 95%CI 12.0–17.4%) (Chi^2^ = 9.081, df = 1, *p* = 0.003). There were 27 *S. aureus* isolates cultured from HIV-positive workers (26 MSSA and 1 MRSA) and 99 *S. aureus* isolates from HIV-negative individuals (97 MSSA and 2 MRSA). There was no difference in detection of MRSA in the two groups (HIV-negative 2/99, 2.0% versus HIV-positive 1/27, 3.7%; Chi^2^ = 0.263, df = 1, *p* = 0.608). There was no significant difference between the proportion of HIV-positive workers (17/89, 19.1%) and HIV-negative workers (110/647, 17.0%) who had recently taken antibiotics (Chi^2^ = 0.241, df = 1, *p* = 0.623).

### 2.3. Genetic Diversity of MSSA and MRSA STs in HIV-Positive and HIV-Negative Participants

Multi-locus sequence typing from the whole genome sequencing of the 126 *S. aureus* isolates identified eleven clonal complexes, consisting of CC1, CC5, CC8, CC15, CC22, CC25, CC30, CC72, CC80, and CC88 (Figure 1). The largest cluster of isolates was CC152. Two ST types accounted for approximately half of all isolates with ST152 (43/126, 34.1%), and ST8 (19/126, 15.1%,) most frequently identified, followed by ST72 (9/126, 7.1%), ST80 (7/126, 5.6%) and ST22 (5/126, 4%). The three MRSA isolates were all ST88. 

HIV-positive participants carried 8 different STs and HIV-negative participants carried 23 different STs, indicating roughly equivalent genetic diversity in the two groups. There was no difference in the carriage of ST152 in HIV-negative (35/99, 35.4%) versus HIV-positive participants (8/27, 29.6%; Chi^2^ = 0.315, df = 1, *p* = 0.575). ST8 was isolated at a higher proportion from HIV-positive (11/27, 40.7%) than HIV-negative participants (8.1%, 8/99) (Chi^2^ = 17.460, df = 1, *p* < 0.001). All ST72 were recovered from HIV-negative participants (Appendix A). 

### 2.4. Prevalence of PVL Gene and Toxic Shock Syndrome Toxin-1 (TSST-1) Gene Carriage

Almost half of isolates (54/126, 42.9%) were *pvl*-positive. There was no difference in prevalence of *pvl* gene carriage between isolates from HIV-positive workers (10/27, 37.0%) and isolates from HIV-negative participants (44/99, 44.4%; Chi^2^ = 0.471, df = 1, *p* = 0.493). The majority of *pvl*-positive isolates were ST152 (39/54, 72.2%). Other STs, with *pvl* gene carriage were ST1633 (4/54, 7.4%), ST30 (3/53, 5.6%), ST88 (2/54, 3.7%), ST80 (2/54, 3.7%), ST2430 (1/54, 1.9%), ST22 (1/54, 1.9%), and ST5 (1/54, 1.9%). The *tsst-1* gene was identified in 15/126 isolates (11.9%), the majority of which were ST 72 (9/15, 60%). The remainder were ST22 (3/15, 20%), ST707 (2/15, 13.3%) and ST8 (1/15, 6.7%). The majority of *tsst-1* positive isolates (*n* = 14) were detected in HIV-negative participants. The three MRSA isolates were negative for both genes.

### 2.5. Prevalence of Phenotypic Antibiotic Resistant S. aureus Carriage in Participants

Antibiotic susceptibility testing of the 126 *S. aureus* isolates using the VITEK 2 instrument demonstrated that all isolates were susceptible to chloramphenicol, daptomycin, fusidic acid, mupirocin, nitrofurantoin, rifampicin, teicoplanin, tigecycline, vancomycin, and clindamycin. Resistance was very low to cefoxitin (2/126, 1.6%), ciprofloxacin (2/126, 1.6%), erythromycin (4/126, 3.2%), gentamicin (2/126, 1.6 %), linezolid (1/126, 0.8 %), oxacillin (1/126, 0.8%), and inducible resistance to clindamycin (4/126, 3.2%). Higher levels of resistance were detected to penicillin-G (123/126, 97.6%); trimethoprim (81/126, 64.3%), tetracycline (33/126, 26.2%) by VITEK 2, and sulfamethoxazole/trimethoprim by disc diffusion (16/126, 12.7%) (Table 2).

Resistance to trimethoprim was not significantly different between isolates from HIV-positive (21/27, 77.8%) and HIV negative (60/90, 60.6%) participants (Chi^2^ = 2.712, df = 1, *p* = 0.100). In contrast, resistance to sulfamethoxazole/trimethoprim by disc diffusion was more common in HIV-positive participants (10/27, 37.0%) compared with HIV-negative participants (6/99, 6.1%, Chi^2^ = 18.098, df = 1, *p* < 0.001). Most sulfamethoxazole-trimethoprim resistance isolates were ST8 (12/16, 75%), others being ST80, ST25 and ST152. Of the sixteen sulfamethoxazole/trimethoprim resistant isolates only one was *pvl*-positive and none carried the *tsst-1* gene.

### 2.6. Prevalence of MDR S. aureus

A total of 34/126 (27.0% (95%CI 20.0–35.3%) isolates were multidrug resistant from the VITEK 2 results. The proportion of MDR isolates was higher in isolates from HIV-positive (9/27, 33.3%) than in isolates from HIV-negative participants (25/99, 25.3%), although this was not statistically significant (Chi^2^ = 0.683, df = 1, *p* = 0.409). The most common resistance combination was trimethoprim, penicillin, and tetracycline, observed in 82.4% (28/34) of MDR isolates and accounting for 22.2% (28/126) of all *S. aureus* isolates (Figure 1). There was no significant difference between isolates demonstrating the resistance pattern of trimethoprim, penicillin and tetracycline from HIV-negative (8/27, 29.6%) versus HIV-positive participants (20/99, 20.2%) (Chi^2^ = 1.076, df = 1, *p* = 0.230). There was no significant difference in the proportion of carriage of MDR isolates between groups with recent and no recent antibiotic use (8/25, 32.0% versus 26/101, 25.7%) (Chi^2^ = 0.401, df = 1, *p* = 0.527).

MDR isolates were predominantly ST8 (26.5%, 9/34) and ST80 (7/34, 20.6%). MDR isolates were less frequently *pvl* positive (9/34, 26.5%) than non-MDR isolates (45/92, 48.9%) (Chi^2^ = 5.046, df = 1, *p* = 0.025). Of the MDR isolates, 8.8% (3/34) were positive for *tsst-1* compared to 13.0% (12/92) of the non-MDR isolates (Chi^2^ = 0.415, df = 1, *p* = 0.519).

Highly resistant MDR *S. aureus* that were resistant to at least five antibiotic classes were isolated from one HIV-positive and three HIV-negative participants.

### 2.7. Prevalence of Genotypic Antibiotic Resistance in MSSA and MRSA Strains

The most common resistance genes detected were beta-lactamase resistance gene *blaZ* (123/126, 96.9%); trimethoprim resistance gene *dfrG* (78/126, 61.9%), and tetracycline resistance gene *tetK* (32/126, 25.4%). Other resistance genes detected included *dfrA* (4/126, 3.2%), *tetM* (1/126, 0.8%), erythromycin resistance gene *ermC* (4/126, 3.2%), and gentamicin resistance gene *aacA-aphD* (2/126, 1.6%). HIV-positive individuals were significantly more frequently colonized by *S. aureus* harboring *dfrA* and *dfrG* genes (22/27, 81.5%) compared to HIV-negative persons (60/99, 60.6%; Chi^2^ = 4.045, df = 1, *p* = 0.044). There was no significant difference prevalence of *tetK* and *tetM* genes between HIV-positive participants (9/27, 33.3%) compared to HIV-negative persons (24/99, 24.2%; (Chi^2^ = 0.902, df = 1, *p* = 0.342).

There was almost 100% agreement between phenotypic and genotypic resistance patterns for benzylpenicillin (121/123, 98.4%), trimethoprim (81/82, 98.8%), tetracycline (31/33 93.9%), erythromycin (4/4, 100%), and gentamicin (2/2, 100%). However, there was discrepancy between the MRSA isolates identified by the presence of *mecA* gene (*n* = 3) and those demonstrating oxacillin resistance (*n* = 1/3, 33.3%) and cefoxitin resistance (*n* = 2/3, 66.6%).

## 3. Discussion

This study described the nasal carriage of *S. aureus* in abattoir workers in western Kenya. The overall nasal carriage of *S. aureus* in this population was 16.0% where 15.6% were MSSA and 0.4% were MRSA. Humans are asymptomatically colonized with nasal *S. aureus* in the range of between 20–30% [40] with MRSA colonization varying between studies and dictated by the methodology used [41]. The carriage of *S. aureus* in this population was lower than expected but consistent with another study conducted in abattoir workers in Nigeria where the prevalence of *S. aureus* carriage was 13.5% [42].

The study population was made up of two groups: HIV-positive abattoir workers and HIV-negative abattoir workers which had an impact on the phenotypes and genotypes of *S. aureus* isolates in this population. The prevalence of HIV infection in this population (12%) was higher than the national average (5%). The reasons for the increased HIV positivity in this population may be related to the sociodemographic group but this was not explored further in this study. The prevalence of *S. aureus* nasal carriage of HIV-positive abattoir workers was significantly higher, 27.0%, when compared to HIV-negative abattoir workers, 14.5%. HIV infection is considered a risk factor for *S. aureus* colonization [22]. This difference in nasal carriage has been reported in previous studies in Africa. In Lagos, Nigeria, HIV-positive study participants were more likely to be colonised with *S. aureus* (33%) compared to HIV-negative participants (21%) [43]. These findings in the sub-Saharan region, suggest that HIV individuals are predisposed to *S. aureus* nasal colonization. Nasal colonization can lead to opportunistic infection in immunocompromised people, and the infection can be life-threatening if not treated promptly [2,44]. This highlights the need to monitor AMR in this population to determine treatment options and improve antimicrobial stewardship.

There was a high proportion of toxigenic strains of *S. aureus* carrying the *pvl* gene (42.9%) and the *tsst-1* gene (11.9%). This is consistent with other studies conducted in sub-Saharan Africa where the carriage of *pvl* genes (33%) is higher than that reported in Europe (5%) [22]. The majority of *pvl* positive strains were ST152-MSSA, which is one of the predominant *pvl*-positive clones in Africa [22,36,45]. The carriage of *pvl*-positive *S. aureus* puts abattoir workers at risk of opportunistic deep skin and soft tissue infections [36,46,47]. The carriage of *S. aureus pvl*-positive strains was not significantly different between HIV-positive abattoir workers (37.0%) and HIV-negative abattoir workers (44.4%) which is consistent with reports from Nigeria, where the proportion of *S. aureus pvl*-positive strains were evenly distributed between isolates from HIV-positive and HIV-negative individuals [43].

There was an inverse relationship between *pvl* carriage and AMR, with 26.5% of MDR isolates being *pvl* positive compared to non-MDR isolates (48.9%) (*p* = 0.025). This differs from other studies in the region where *pvl* carriage has been associated with MDR. An association between *pvl* carriage and sulfamethoxazole/trimethoprim resistance has been observed in Gabon and Nigeria among HIV-positive individuals [43,46].

The prevalence of MRSA carriage (0.4%) identified in this study was low compared to studies conducted with abattoir workers in Europe (5.6%) [48], and the USA 3.6% [16]. However, our results are consistent with other reports of MRSA carriage in Kenya (0.8%) [26] but much lower than studies of MRSA cultured from hospital patients in Kenya (53.4%) [49]. This may suggest that MRSA infection in Kenya is predominantly linked to the hospital environment rather than acquired from the community. The three MRSA isolates belonged to ST88, which is referred to as the “African” community-associated (CA-MRSA) clone [50] but is not the most reported MRSA sequence type in Kenya, which is ST239 [24]. ST88 has been reported in pigs and workers in a Nigerian abattoir and may indicate an animal source [51]. Antimicrobial use in animals is not regulated in Kenya and the most frequently used antibiotics in animals in western Kenya are oxytetracycline and penicillin-streptomycin [52]. Further work to understand MRSA carriage in animals, as well as spread in the human population and environment, is required.

The findings from this study supported prior evidence of *S. aureus* resistance to penicillin, tetracycline, and sulfamethoxazole/trimethoprim in the sub-Saharan region [22]. The proportion of isolates that were phenotypically resistant to penicillin (97.6%), tetracycline (26.2%), and sulfamethoxazole/trimethoprim (12.7%) was consistent with previous studies from Kenya reporting marked resistance to penicillin (76–100%) and moderate resistance to tetracycline (15–20%) and sulfamethoxazole/trimethoprim (30–40%) [25,26]. There was increased frequency of sulfamethoxazole/trimethoprim resistance in HIV-positive abattoir workers (37%) compared to HIV-negative workers (6.1%) (*p* < 0.001), with the majority of resistant strains belonging to ST8-MSSA.

There was high genotype-phenotype concordance between resistance genes detected and antimicrobial susceptibility test (AST) results for most antimicrobials as has been previously reported [25]. However, there were two *mecA* positive MRSA strains that were susceptible to oxacillin and one of these was also susceptible to cefoxitin. This may be due to a misclassification error, although the sensitivity of the VITEK 2 instrument for detecting oxacillin resistance is high (97.5%) [53]. Alternatively, this may indicate oxacillin susceptible *mecA* MRSA (OS-MRSA) strains are circulating in this environment. OS-MRSA strains have previously been identified in other parts of Africa mainly associated with ST88 as described here [54]. Information regarding the presence of the *mecC* gene in these isolates was not available. The presence of OS-MRSA in this setting may complicate treatment options that are based solely on AST results, since OS-MRSA may be misidentified as MSSA, and these isolates can develop β-lactam resistance following antibiotic therapy [55].

Sulfamethoxazole/trimethoprim is an effective antibiotic combination in the treatment and prevention of bacterial infections in people who are HIV positive and has been used to treat *Pneumocystis jiroveci* pneumonia and other bacterial infections in severely immunocompromised HIV-positive individuals. The prophylactic use of sulfamethoxazole/trimethoprim in all HIV-positive individuals regardless of CD4 counts, especially in regions having high prevalence of malaria and/or severe bacterial infections, such as sub-Saharan region [56], may have resulted in the high prevalence of sulfamethoxazole/trimethoprim resistant *S. aureus* in Africa [43,46]. This, coupled with the extensive use of penicillin and tetracycline for use in animal production for both therapeutic and non-therapeutic purposes in Africa [12], has created a favorable environment for the emergence of multidrug resistance *S. aureus* through antibiotic related selective pressure [57,58]. These MDR *S. aureus* reduce the treatment options for effective treatment for HIV-positive individuals with opportunistic infections. With the presence of MDR *S. aureus* that are resistant to additional multiple resistant combinations, including erythromycin, clindamycin, ciprofloxacin and gentamicin, the treatment options will be further diminished and become more expensive. The improved availability of antiretroviral therapy (ART) in sub-Saharan Africa, has led to reduction of severely immune compromised HIV-positive individuals and fewer cases of serious *Pneumocystis jiroveci* pneumonia and opportunistic infections among HIV-positive individuals [59]. Improved health outcomes for HIV-positive individuals through access to ART may lead to reduced prophylactic use of sulfmethoxazole/trimethoprim in this region. This will aid in the preservation of therapeutic advantages of this affordable drug in treatment and prevention of bacterial infections.

Multi-drug resistant *S. aureus* strains colonizing HIV-positive abattoir workers may be a risk for SSTI since 25% of workers reported being injured at work [60]. MDR *S. aureus* could also be transmitted to the community directly or through the meat supply chain to consumers [4]. In addition, if the respective abattoir workers are hospitalized, the strains can be spread to hospital staff and compromised inpatients, threatening effective treatment of resultant infections [61]. Thus, MDR *S. aureus*-colonized abattoir workers pose a public health problem in hospitals, community, and food industry in Busia County, which can extend to other parts of Kenya and neighboring countries of Kenya since Busia is a border town.

There is a need for increased monitoring of antibiotic usage and surveillance measures for AMR bacteria in both animals and humans in this region already burdened by HIV/AIDS infection and where there is rapidly increasing demand for meat products caused by population growth and urbanization [12]. Additionally, there is a need for strategies to promote the prudent use of antimicrobials and antimicrobial stewardship as described by the global strategy [62]. It is particularly important to strategize on the appropriate use of sulfamethoxazole/trimethoprim, tetracyclines and penicillin in human and animal healthcare and food production in sub-Saharan region, since there were high proportions of resistant isolates to these antimicrobials. It has been demonstrated that a reduction of antibiotic consumption leads to decreased prevalence of antimicrobial resistance [63]. This can be done through national action plans for the prevention and containment of AMR with contributions from human and animal health agencies [39].

There was a delay between the collection of samples and the publication of these findings which may limit the usefulness of these results. However, given the lack of available data regarding the circulating MSSA and MRSA strains and antimicrobial resistance profiles of *S. aureus* in this population the data is a valuable contribution to knowledge regarding AMR in the region and may prove a useful baseline for comparison to future studies. Data regarding the circulating MSSA and MRSA strains in livestock was not available at the time of this publication. This information would have been useful to understand the potential for transmission of isolates between livestock and workers and is a data gap that should be targeted in future research.

## 4. Materials and Methods

### 4.1. Study Site

The study area was a 45 km radius from Busia town including most of Busia County, and parts of Bungoma, Siaya and Kakamega Counties spanning the 3200 km^2^ of the Lake Victoria ecosystem. The predominant industry in the study area is subsistence agriculture [64]. A census of all abattoirs (*n* = 156) was conducted in the study area in 2012. Fourteen abattoirs declined to participate. Participants were recruited from 142 abattoirs, 84 ruminants and 58 porcine [60].

### 4.2. Study Population

A total of 738 abattoir workers were recruited into the study between February and November 2012 from a total of 1005 workers (73.3%) in the selected slaughterhouses. In abattoirs with 12 workers or less all consenting workers were recruited and in abattoirs with more than 12 workers a random selection of twelve workers were recruited [60].

### 4.3. Data and Sample Collection

All participants were informed of the project objectives and protocol by a clinical officer who collected signed informed consent. Data was collected using a structured questionnaire regarding demographic details, health events and recent antimicrobial use. Nasal samples were collected by rotating a sterile swab five times in both anterior nares, from consenting abattoir workers. The swabs were inoculated in tryptone soya broth with 6% salt and transported in cool boxes to the laboratory for culturing.

Blood was collected by a clinical officer into 4 mL EDTA vacutainers using a butterfly catheter (BD Vacutainer^®^ Safety-Lok™ blood collection set). Samples were transported in cool boxes to the laboratory in Busia. Whole blood samples were stored frozen at −40 °C until transportation to the International Livestock Research Institute (ILRI) laboratory in Nairobi for long term storage at −80 °C. HIV testing was performed on whole blood using the SD Bioline HIV 1/2 Fast 3.0 test strips (Standard Diagnostics Inc., Suwon-si, South Korea).

Swabs were streaked onto Mannitol Salt agar and incubated at 37 °C in air overnight. Suspect *S. aureus* isolates, those fermenting mannitol and producing yellow colonies, were stocked in tryptone soya broth with 10% glycerol and stored at −40 °C and transported on dry ice to Nairobi. Presumptive *S. aureus* isolates were further cultured onto mannitol salt agar (MSA) and sub-cultured to obtain pure culture. The *S. aureus* isolates were identified using Gram reaction (Gram-positive cocci in clumps), catalase, coagulase (tube method using rabbit plasma) and DNase tests. Initially, a long sweep of the colonies was done to allow preservation of genetic diversity of nasal carriage of the participant. During whole genome sequencing, some samples were shown to consist of mixed isolates and these samples were recultured to select single colonies for sequencing and antimicrobial susceptibility testing.

### 4.4. Phenotypic Antimicrobial Susceptibility Testing

Antimicrobial susceptibility testing (AST) was performed using the VITEK 2 instrument (bioMerieux, Marcy-l’Étoile, France), for 20 antimicrobials including: benzylpenicillin, cefoxitin, oxacillin, ciprofloxacin, erythromycin, chloramphenicol, daptomycin, fusidic acid, gentamicin, linezolid, mupirocin, nitrofurantoin, rifampicin, teicoplanin, tetracycline, tigecycline, trimethoprim, vancomycin, clindamycin, and inducible resistance to clindamycin. The Vitek 2 system uses fluorescence, turbidity and colormetric methods to monitor bacterial growth and uses this to calculate minimum inhibitory concentrations (MICs) [65].

Additionally, due to the clinical importance, antimicrobial resistance to sulfamethoxazole/trimethoprim was tested using the Kirby-Bauer disc diffusion using Clinical and Laboratory Standards Institute (CLSI) guidelines (1.25 + 23.75 µg; TMP/SXT). Zone diameter interpretive standards were: sensitive ≥16 mm, intermediate 11–15 mm and resistant 10 ≤ [66].

Multi-drug resistant *S. aureus* were defined as isolates that were resistant to three or more different antimicrobials using the VITEK results [67,68,69]. Isolates resistant to 5 or more antimicrobials were described as highly multi-drug resistant.

### 4.5. Molecular Genotyping

DNA extraction from *S. aureus* isolates was performed using QIAGEN DNeasy Blood & Tissue kit (QIAGEN, Valencia, CA, USA). Staphylococcal cassette chromosome (SCC) *mec* typing were performed using previously described methods [70]. Isolates with the *mecA* gene were classified as MRSA. Panton-Valentine Leukocidin (*pvl*) and toxic shock syndrome toxin-1 (*tsst-1*) gene detection was done by PCR using previously described oligonucleotide primers [71,72] at the Kenya Medical Research Institute Laboratories (KEMRI), Nairobi, Kenya.

DNA extraction from *S. aureus* isolates was performed on a QIAcube, using the QIAamp 96 HT kit (QIAGEN). Genomic libraries were generated and sequenced on an Illumina HiSeq 2000 (Illumina Inc., San Diego, CA, USA) at the Wellcome Sanger Institute, Hinxton, UK. Illumina reads were analysed based on the *S*. *aureus* MLST database ( https://pubmlst.org/organisms/staphylococcus-aureus, accessed on 31 March 2021) [73], and analysis of virulence and antimicrobial resistance genes were conducted using virulence finder database (https://cge.cbs.dtu.dk/services/, accessed on 31 March 2021).

### 4.6. Genomic Analyses

Paired-end Illumina reads were mapped to the *S. aureus* reference genome ST22 strain HO 5096 0412 (accession number HE681097) using Snippy v4.6.0 (https://github.com/tseemann/snippy, accessed on 31 October 2022). Whole-genome alignments were created by keeping a version of the reference genome with only substitution variants replaced (i.e., SNPs but not indels) using Snippy’s *.consensus.subs.fa* output files. The *S. aureus* species core-genome had been previously derived [74] from a collection of 800 *S. aureus* from multiple host species [75]. The portion of the reference genome (2.83 Mb) corresponding to the core genome (1.76 Mb) was kept from whole-genome alignments and used to generate maximum likelihood trees using IQ-TREE v1.6.10 with default settings. The resulting core-genome phylogeny was plotted with isolate metadata using ggtree v.3.0.4 [76] and ggtreeExtra v.1.2.3 on R v4.1.0 [77].

### 4.7. Statistical Analyses

Statistical analysis was performed using the chi-squared test. A *p*-value <0.05 was considered an indication of significant difference.

### 4.8. Ethical Approval

The study was approved by the Centre for Microbiology Research Centre Scientific Committee, Kenya Medical Research Institute scientific steering Committee and Ethical Review Committee (SSC No 2086 granted 31 October 2011 and 2944 granted 13 May 2015).

## 5. Conclusions

This study identifies the circulating MSSA and MRSA strains in a population occupationally exposed to livestock in rural western Kenya. This gives an improved understanding of the epidemiology of *S. aureus* particularly the strains, sources, and risk groups in a setting that has not been previously studied. More importantly, the study indicates the levels of AMR and prevalence of toxigenic genes in *S. aureus* isolates, which is particularly important in this community with high prevalence of immunocompromised individuals. This information can contribute to developing measures for the prevention and containment of AMR.

## Figures and Tables

**Figure 1 antibiotics-11-01726-f001:**
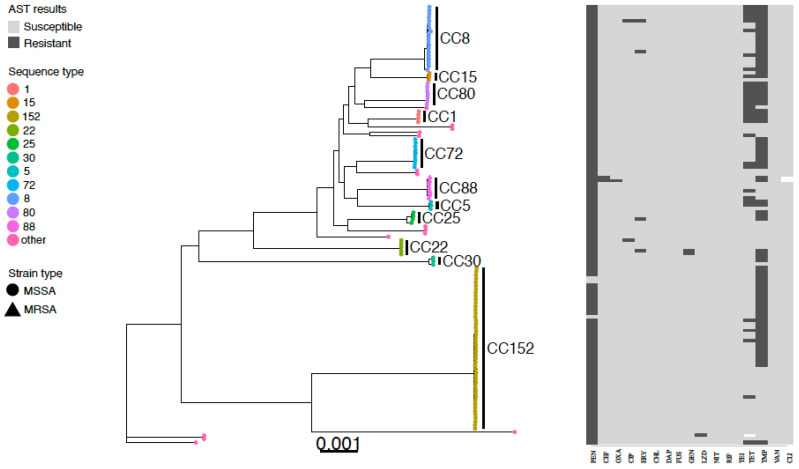
Core genome phylogenetic tree and antibiogram of MSSA and MRSA isolates colonizing abattoir workers in western Kenya. Predominant ST types are differentiated by colour, MRSA isolates are indicated by a triangle symbol and MSSA by circles. Phenotypic antimicrobial resistance is indicated by dark grey bars (resistant), white bars (intermediate), and light grey (susceptible). Antimicrobials: PEN—benzylpenicillin, CEF—cefoxitin, OXA—oxacillin, CIP—ciprofloxacin, ERY—erythromycin, CHL—chloramphenicol, DAP—daptomycin, FUS—fusidic acid, GEN—gentamicin, LZD- linezolid, NIT—nitrofurantoin, RIF—rifampicin, TEI—teicoplanin, TET—tetracycline, TMP—trimethoprim, VAN—vancomycin, CLI—clindamycin.

**Table 1 antibiotics-11-01726-t001:** Prevalence of MSSA and MRSA isolated from Abattoir workers in Busia County.

HIV-Status	Number of *S. aureus* Isolates (Number of MRSA)	Number of Workers Who Yielded *Staph. aureus* (%, 95% Confidence Interval)	Number of Workers Carrying MRSA Isolates (%, 95%CI)	Number of Workers Carrying MSSA Isolates (%, 95%CI)
HIV-positive (*n* = 89)	27 (1)	24 (27.0%, 95%CI 18.9–37.1)	1 (1.1%, 95%CI 0.3–6.0)	23 (25.8%, 95%CI 18.0–26.0)
HIV-negative (*n* = 648)	99 (2)	94 (14.5%, 95%CI 11.9–17.4)	2 (0.3%, 95%CI 0.1–1.1)	92 (14.2%, 95%CI 11.7–17.1)
Total abattoir workers (*n* = 737)	126 (3)	118 (16.0%, 95%CI 13.5–29.0)	3 (0.4%, 95%CI 0.1–1.2)	115 (15.6%, 95%CI 13.1–18.3)

**Table 2 antibiotics-11-01726-t002:** Prevalence of antibiotic resistant *Staph. aureus* carriage among HIV-positive and HIV-negative abattoir workers in Busia County.

	Total Number of Isolates *n* = 126 (%)	Isolates from HIV-Positive Workers *n* = 27 (%)	Isolates from HIV-Negative Workers *n* = 99 (%)	Chi^2^ Test
Sulfamethoxazole/trimehoprim resistance	16 (12.7)	10 (37.0)	6 (6.1)	Chi^2^ = 18.098, df = 1, *p* <0.001
Penicillin	123 (97.6)	26 (100)	96 (97.0)	NA
Trimethoprim	81 (64.3)	21 (77.8)	60 (60.6)	Chi^2^ = 2.712, df = 1, *p* = 0.100
Cefoxitin	2 (1.6%)	1 (3.7)	2 (2.0)	Chi^2^ = 0.263, df = 1, *p* = 0.608
Tetracycline	33 (26.2)	10 (37.0)	23 (23.2)	Chi^2^ = 2.075, df = 1, *p* = 0.150
Erythromycin	4 (3.2)	2 (7.4)	2 (2.0)	Chi^2^ = 2.007, df = 1, *p* = 0.157
Inducible resistance to clindamycin	4 (3.2)	2 (7.4)	2 (2.0)	Chi^2^ = 2.007, df = 1, *p* = 0.157
Gentamicin	2 (1.6)	0	2 (2.0)	NA
Ciprofloxacin	2 (1.6)	2 (7.4)	0	NA
Oxacillin	1 (0.8)	0	1 (1.0)	NA
Linezolid	1 (0.8)	0	1 (1.0)	NA
Panton Valentine Leukocidin gene	54 (42.9)	10 (37.0)	44 (44.4)	Chi^2^ = 0.471, df = 1, *p* = 0.493
Toxic shock syndrome toxin-1 gene	15 (11.9)	1 (3.7)	14 (14.1)	Chi^2^ = 2.176, df = 1, *p* = 0.140
Multidrug-resistant	34 (27.0)	9 (33.3)	25 (25.3)	Chi^2^ =0.683, df = 1, *p* = 0.409
Trimethoprim-Penicillin-Tetracycline MDR	28 (22.2)	8 (29.6)	20 (20.2)	Chi^2^ = 1.076, df = 1, *p* = 0.230

## Data Availability

Data is contained within the article and Appendix A.

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
