# Peer review of "Multi-Drug Resistant Staphylococcus aureus Carriage in Abattoir Workers in Busia, Kenya"

_antibiotics, 2022, doi:10.3390/antibiotics11121726_

Round 1

Reviewer 1 Report

I have very few suggestions on how to improve the manuscript:

1.     Lines 7\9\14…. – there are no country names

2.     there is no need to skip a line between paragraphs in the abstract

3.     «S. aureus was 27 isolated using culture» - what culture? What do you mean? Write “culture technique” or something like this

4.     There is no need to number keywords

5.     line 99 «S. 98 aureus was isolated from 118/737» – maybe 738? Or in line 96 should be 737?

6.     Figure 1 – “TMP” in the image, but “TRI” in the description

7.     I don't quite understand why abattoir workers with HIV were chosen. Yes, there is a problem of uncontrolled use of antibiotics in animal husbandry. There is also the problem of immunosuppression and more frequent bacterial infections in HIV patients. Describe the relationship of these things in more detail, perhaps in the form of a graphic diagram. There is a connection but improve illustration of this idea.

Author Response

Thank you for your review. We have addressed your comments in the manuscript and below.

I have very few suggestions on how to improve the manuscript:

  1. Lines 7\9\14…. – there are no country names

This has been corrected and country names added

  1. there is no need to skip a line between paragraphs in the abstract

The space has been removed

  1. «S. aureus was 27 isolated using culture» - what culture? What do you mean? Write “culture technique” or something like this

We have added “bacterial culture” to indicate the type of culture

  1. There is no need to number keywords

These numbers have been removed

  1. line 99 « 98 aureus was isolated from 118/737» – maybe 738? Or in line 96 should be 737?

This has been reworded for clarity

  1. Figure 1 – “TMP” in the image, but “TRI” in the description

            This has been corrected

  1. I don't quite understand why abattoir workers with HIV were chosen. Yes, there is a problem of uncontrolled use of antibiotics in animal husbandry. There is also the problem of immunosuppression and more frequent bacterial infections in HIV patients. Describe the relationship of these things in more detail, perhaps in the form of a graphic diagram. There is a connection but improve illustration of this idea.

We have reworded and added details to the introduction separating the risks to abattoir workers and indicating why this population was valuable for the comparison between HIV positive and HIV negative individuals.  

Reviewer 2 Report

The manuscript entitled "Multi-drug resistant Staphylococcus aureus carriage in abattoir workers in Busia, Kenya" aimed to investigate the association between HIV infection and S. aureus in abattoir workers in Western Kenya. The study has an importance in the field of antimicrobial resistance and HIV infection. However, the manuscript requires few corrections, English editing is also required. The corrections are given bellow.

Line no. 25: Authors have stated that, "This study investigated the association between HIV infection and S. aureus in abattoir workers in Western Kenya." However, I think, they may focus "HIV" in title too.

Line no. 32-33: Please rewrite the sentence "There were 23 sequence types (STs), the dominant being ST152 (34.1%) and ST8 (15.1%)."

Introduction section should be improved.  HIV, S. aureus and resistance interactions are lees focused. In addition, livestock-associated Staphylococcus aureus (LA-SA), its importance for infection to abattoir workers, and spread to community have not been discussed well.

Section 2.1 (Prevalence of MRSA and MSSA among HIV-positive and HIV-negative participants) is written based on table no. 1. However, from line no. 99 to 103, these data are not found in table no. 1.  Moreover, line no. 105-115 is showing the same problem. Authors are showing the prevalence based on worker only in table 1. But, when describing they are using both the prevalence based on worker and S. aureus. This will create confusion for the readers.  I am requesting the Authors to add the prevalence based on S. aureus isolates as well. They may also add addition column for 95% CI for both the prevalence in the same table.

Line no. 231-232: Please rewrite the sentence.

Line no. 275: Please elaborate AST.

Discussion section may have a limitation part of the studies.

What is the basis of taking 738 samples? Did the Authors use any statistical sample calculation method?

Line no. 355: Please add few sentences on how the VITEK 2 instrument works for Antimicrobial susceptibility testing and its interpretation criteria with references.  

Conclusion section should be improved. This section should have the main findings in short. Then the Authors may add their recommendations and/or own think.  

Author Response

Thank you for your review. We have copied your comments below and responded point-by-point as well as making changes to the manuscript. 

The manuscript entitled "Multi-drug resistant Staphylococcus aureus carriage in abattoir workers in Busia, Kenya" aimed to investigate the association between HIV infection and S. aureus in abattoir workers in Western Kenya. The study has an importance in the field of antimicrobial resistance and HIV infection. However, the manuscript requires few corrections, English editing is also required. The corrections are given bellow.

We have extensively edited for English language, particularly the introduction.

Line no. 25: Authors have stated that, "This study investigated the association between HIV infection and S. aureus in abattoir workers in Western Kenya." However, I think, they may focus "HIV" in title too.

We think that the title captures enough detail while remaining succinct and to the point. The comparison between HIV positive and HIV negative workers is one risk factor that was examined within the population. We have amended the aim to better reflect the purpose of the study to investigate S. aureus in abattoir workers.

Line no. 32-33: Please rewrite the sentence "There were 23 sequence types (STs), the dominant being ST152 (34.1%) and ST8 (15.1%)."

This has been reworded in the abstract

Introduction section should be improved.  HIV, S. aureus and resistance interactions are lees focused. In addition, livestock-associated Staphylococcus aureus (LA-SA), its importance for infection to abattoir workers, and spread to community have not been discussed well.

We have rewritten the introduction and added further details to the introduction separating the risks to abattoir workers and their role in transmission and then indicating why this population was valuable for the comparison between HIV positive and HIV negative individuals. 

Section 2.1 (Prevalence of MRSA and MSSA among HIV-positive and HIV-negative participants) is written based on table no. 1. However, from line no. 99 to 103, these data are not found in table no. 1.  Moreover, line no. 105-115 is showing the same problem. Authors are showing the prevalence based on worker only in table 1. But, when describing they are using both the prevalence based on worker and S. aureus. This will create confusion for the readers.  I am requesting the Authors to add the prevalence based on S. aureus isolates as well. They may also add addition column for 95% CI for both the prevalence in the same table.

This has been added to the table and further clarification added to the text

Line no. 231-232: Please rewrite the sentence.

This has been done - In Lagos, Nigeria, HIV-positive study participants were more likely to be colonised with S. aureus (33%) compared to HIV-negative participants (21%) (43)

Line no. 275: Please elaborate AST.

This has been done

Discussion section may have a limitation part of the studies.

This has been added

“There was a delay between the collection of samples and the publication of these findings which may limit the usefulness of these results. However, given the lack of available data regarding the circulating MSSA and MRSA strains and antimicrobial resistance profiles of S. aureus in this population the data is a valuable contribution to knowledge regarding AMR in the region and may prove a useful baseline for comparison to future studies. Data regarding the circulating MSSA and MRSA strains in livestock was not available at the time of this publication. This information would have been useful to understand the potential for transmission of isolates between livestock and workers and is a data gap that should be targeted in future research.”  

What is the basis of taking 738 samples? Did the Authors use any statistical sample calculation method?

We had initially planned to do a census but due to the limitations on the number of samples that could be processed per day we reduced the sample size. The number of workers in the recruited slaughterhouses was 1005. We managed to recruit 737 workers which is 73% and should be representative of the population. We have added this to the methods

Line no. 355: Please add few sentences on how the VITEK 2 instrument works for Antimicrobial susceptibility testing and its interpretation criteria with references.  

This has been added “The Vitek 2 system uses fluorescence, turbidity and colormetric methods to monitor bacterial growth and uses this to calculate minimum inhibitory concentrations (MICs) (62).”

Conclusion section should be improved. This section should have the main findings in short. Then the Authors may add their recommendations and/or own think.  

This has been done “This study identifies the circulating MSSA and MRSA strains in a population occupationally exposed to livestock in rural western Kenya. More importantly the study indicates the levels of AMR and prevalence of toxigenic genes in S. aureus isolates, which is particularly important in this community with high prevalence of immunocompromised individuals. There is a need for increased surveillance measures and monitoring of antibiotic usage in both animals and humans in this region already burdened by HIV/AIDS infection and where there is rapidly increasing demand for meat products caused by population growth and urbanization (12).

There is a need for prudent use of antimicrobials and antimicrobial stewardship as described by the global strategy (74). It is particularly important to strategize on the appropriate use of sulfamethoxazole/trimethoprim, tetracyclines and penicillin in human and animal healthcare and food production in sub-Saharan region, since there were high proportions of resistant isolates to these antimicrobials. It has been demonstrated that a reduction of antibiotic consumption leads to decreased prevalence of antimicrobial resistance (75). This can be done through national action plans for the prevention and containment of AMR with contributions from human and animal health agencies (39).”

Round 2

Reviewer 1 Report

Conclusions should reflect the main idea of the manuscript, so you should not focus on what should be done in the future. Please, do not give references in conclusions, move the references to the discussion.

Author Response

Dear Reviewer

Many thanks for your continued efforts to improve our manuscript. 

We have reworded the conclusion, and moved the additional information to the discussion. 

The conclusion now reads

This study identifies the circulating MSSA and MRSA strains in a population occupationally exposed to livestock in rural western Kenya. This gives an improved understanding of the epidemiology of S. aureus particularly the strains, sources, and risk groups in a setting that has not been previously studied. More importantly, the study indicates the levels of AMR and prevalence of toxigenic genes in S. aureus isolates, which is particularly important in this community with high prevalence of immunocompromised individuals. This information can contribute to developing measures for the prevention and containment of AMR.